# Interleukin-31 and Pruritic Skin

**DOI:** 10.3390/jcm10091906

**Published:** 2021-04-28

**Authors:** Masutaka Furue, Mihoko Furue

**Affiliations:** 1Kyushu University, Higashiku, Fukuoka 812-8582, Japan; 2Independent Scholar, 1-19-20 Momochi, Sawara-ku, Fukuoka 814-0006, Japan; furue1@jcom.home.ne.jp

**Keywords:** IL-31, pruritus, itch, atopic dermatitis, prurigo nodularis, nemolizumab, Th2 cell, macrophage, dendritic cell

## Abstract

Skin inflammation often evokes pruritus, which is the major subjective symptom in many inflammatory skin diseases such as atopic dermatitis and prurigo nodularis. Pruritus or itch is a specific sensation found only in the skin. Recent studies have stressed the pivotal role played by interleukin-31 (IL-31) in the sensation of pruritus. IL-31 is produced by various cells including T helper 2 cells, macrophages, dendritic cells and eosinophils. IL-31 signals via a heterodimeric receptor composed of IL-31 receptor A (IL-31RA) and oncostatin M receptor β. Recent clinical trials have shown that the anti-IL-31RA antibody nemolizumab can successfully decrease pruritus in patients with atopic dermatitis and prurigo nodularis. The IL-31 pathway and pruritic skin are highlighted in this review article.

## 1. Introduction

Pruritus or itch is a specific sensation of the skin leading to a scratching behavior mostly recognized in terrestrial mammals [1,2]. Acute pruritus is typically not a major problem. However, chronic pruritus that lasts more than 6 weeks is highly problematic because it markedly deteriorates quality of life, mental health and sleep quality in the afflicted individuals [1,2,3,4,5]. Chronic pruritus is the major symptom in many inflammatory skin diseases such as atopic dermatitis, prurigo nodularis, eczema, chronic urticaria and psoriasis [1,2,3,4]. Scratching clearly exacerbates the predisposing dermatitis, which may further enhance pruritus and result in an itch/scratch vicious cycle [6].

Recent studies have revealed numerous itch-inducing factors or pruritogens [1,2,3,4,7,8]. Specific receptors for these pruritogens are expressed on cutaneous sensory nerve fibers, which are mostly small C-fibers innervating the epidermis and dermis [9]. Representative pruritogens and their receptors are listed in Table 1.

Among these pruritogens, particular attention has been paid to interleukin-31 (IL-31), since systemic administration of the anti-IL-31 receptor A (IL-31RA) antibody nemolizumab successfully ameliorates chronic itch in patients with atopic dermatitis and prurigo nodularis [10,11].

## 2. IL-31 and Its Signaling

IL-31 belongs to the gp130/IL-6 cytokine family [12]. It was originally reported to be preferentially produced by T helper type 2 (Th2) cells [12]. IL-31 is expressed in Th2, but not in Th1, Th17 and Th22, clones [13]. The production of IL-31 in Th2 clones is dependent on their autocrine IL-4 production [13]. IL-33 strongly potentiates the IL-4-induced IL-31 production in Th2 clones [13]. Th1 clones can produce IL-31 in the presence of IL-4, but only transiently [13].

In addition, IL-31 is expressed by other hematopoietic cells including macrophages (especially M2 macrophages), dendritic cells, mast cells, basophils and eosinophils [12,14,15,16,17,18]. Even epidermal and sweat gland keratinocytes may possibly produce it [19,20,21]. Ultraviolet (UV)B irradiation or oxidative stress upregulates the IL-31 expression in T cells, monocytes, macrophages, and immature and especially mature dendritic cells [21]. Oxidative stress, but not UVB irradiation, also induces IL-31 expression in keratinocytes [21].

The production of IL-31 is upregulated by IL-4-mediated activation of signal transducer and activator of transcription 6 (STAT6) or by IL-33-mediated NF-κB activation [22], while it is downregulated by the activation of suppressor of cytokine signaling 3 (SOCS3) [23]. Staphylococcal enterotoxin B and staphylococcal α-toxin are potent inducers of IL-31 from peripheral blood mononuclear cells, probably via toll-like receptor (TLR)-4 activation [24,25,26]. Interestingly, transforming growth factor-β1 (TGF-β1) decreases IL-31 production in Th2 cells [22], but it enhances that in dendritic cells [18]. Endothelial PAS domain protein 1 (EPAS1), also called hypoxia-inducible factor-2α, is one of the direct targets of STAT6 and acts as a hub protein in IL-4-mediated transcription networks in human T cells [27]. The binding of Epas1 and another transcription factor, Sp1, to the promoter region of the *Il31* gene is indispensable for IL-31 production by IL-4 [28].

The interaction of Epas1 and Sp1 has been reported in hypoxic conditions [29,30]. As the expression of EPAS1 is also upregulated by TGF-β1 [31] and TLR-4 ligation [32], EPAS1 may also potentially be involved in TGF-β1-induced or TLR-4-induced IL-31 upregulation. On the other hand, macrophages express neurokinin 1 receptor (NK1R) (receptor for substance P), CD163 (receptor for hemosiderin) and αV integrin (receptor for periostin) [16]. In macrophages, periostin upregulates IL-31 expression, which is further enhanced by substance P and hemosiderin [16]. This recent evidence suggests that IL-31 expression may be regulated promiscuously by diverse signaling pathways.

IL-31 binds and activates a heterodimeric receptor composed of IL-31 receptor A (IL-31RA) and oncostatin M receptor β (OSMRβ) [12]. The heterodimeric IL-31R is known to be expressed on sensory neurons, macrophages, dendritic cells, basophils and epithelial cells including keratinocytes [12]. Intriguingly, CD3^+^ T cells do not express IL-31RA [33]. IL-4 enhances the expression of IL-31RA in macrophages and dendritic cells [34,35], but not in keratinocytes [36]. Its expression is augmented by interferon-γ or TLR2 ligands in keratinocytes [36,37,38]. IL-31 itself does not affect IL-31RA expression [38]. Stimulation of IL-31RA/OSMRβ by IL-31 activates the downstream Janus kinase 1 (JAK1)/JAK2 and STAT3 (and also to a lesser extent STAT1 and STAT5) signaling pathway [39] (Figure 1).

IL-31 binds to IL-31 receptor (IL-31RA/OSMRβ) and activates JAK1/JAK2 and downstream STAT3 (to a lesser extent STAT1 and STAT5), then induces pruritus.

## 3. IL-31 and Pruritus

The most prominent pathophysiological function of IL-31 is that it is a pruritogenic cytokine in mammals [12]. Despite the low interspecies homology of the *IL31* gene, it induces pruritus and scratch response in rodents, dogs, cynomolgus monkeys and humans [39,40,41,42,43,44,45,46]. Continuous scratching behaviors are observed in IL-31 transgenic mice, which results in the development of atopic dermatitis-like skin lesions [39]. In addition, the administration of IL-31 intravenously, intradermally, subcutaneously or intrathecally induces severe pruritus in normal mice [39,40,41,47,48]. Continuous or repeated injection of IL-31 also evokes sustained scratching and the development of dermatitis [47,48]. Moreover, *Il31ra*-deficient mice do not develop pruritus and dermatitis in response to mouse IL-31 [39], and the administration of an anti-mouse IL-31 antibody ameliorates the scratching behavior in NC/Nga mice with atopic dermatitis-like skin lesions [49]. However, the IL-31-induced pruritus is not significantly attenuated by H1-antihistamine, dexamethasone, tacrolimus or μ-opioid receptor antagonist in mice [41].

As described in previous review articles [7,50,51], the sensation of pruritus in the skin is mediated by cutaneous sensory nerves of small diameter (C-fibers or thinly myelinated Aδ-fibers) originating from dorsal root ganglion (DRG) neuronal cells, and is transmitted to the spinal cord and hypothalamic tract, and ultimately to the brain. Recent studies have demonstrated various pruritogens, pruritogen-specific neurotransmitters and their receptors [7,50,51] (Table 1). BAM8-22, SLIGR and cathepsin S induce pruritus via Mas-related G-protein-coupled receptor C11 (MrgprC11) [9]. Most MrgprC11^+^ cutaneous sensory neurons are positive for substance P and calcitonin gene-related peptide (CGRP) in mice [9] (Figure 2).

The murine MrgprC11^+^ cutaneous sensory neurons also coexpress IL-31RA/OSMRβ, histamine H1 receptor (H1R), MrgprA3 and 5-hydroxytryptamine receptor 1F (5-HTR1F) [9]. The expression of IL-31RA has also been confirmed in cutaneous neurons and DRG in cynomolgus monkeys and humans [45,52]. These human and mouse IL-31RA^+^ neurons express transient receptor potential vanilloid 1 (TRPV1), which is an ion channel for Ca^2+^ and to a lesser extent Na^+^ [9,40]. IL-31-induced itch is significantly reduced in *Trpv1*-deficient mice [40]. IL-31-induced itch is also significantly reduced in mice deficient in *Trpa1*, which is required for itch sensation mediated by MargprA3 [40]. IL-31RA^+^ DRG cells also coexpress IL-4Rα/IL-13Rα1, which is a heterodimeric receptor for IL-4 and IL-13 [53]. IL-4 and IL-13 themselves are pruritogenic cytokines [54]. More importantly, IL-4 and IL-13 further enhance the IL-31-evoked or histamine-evoked action potential in DRG cells, suggesting their participation in chronic itch formation [53].

Many pruritogenic axes such as chloroquine/MrgprA3- or histamine/H1R-induced pruritus require natriuretic polypeptide b (Nppb) and gastrin-releasing peptide (GRP) and their respective receptors, natriuretic peptide receptor A (NPRA) and GRP receptor (GRPR), to transmit the sensation of itch in the spinal cord [55,56] (Figure 2). However, IL-31-induced pruritus requires neurokinin B (NKB) instead of Nppb [56]. NKB binds neurokinin receptor 3 (NK3R) and induces the release of GRP, which mediates the sensation of pruritus [56] (Figure 2).

The majority of Mas-related G-protein-coupled receptor C11 (MrgprC11)-positive murine sensory nerve fibers possess substance P and CGRP. MrgprC11^+^ sensory nerve fibers are positive for TRPV1 and TRPA1 which evokes action potential mainly via Ca^2+^. These MrgprC11^+^ sensory nerve fibers mostly coexpress IL-31 receptor (IL-31RA/OSMRβ), which signals IL-31-induced pruritus to DRG neurons, the spinal cord, the hypothalamic tract and finally to the brain. The IL-31-induced pruritus is mediated by NKB and its receptor NK3R, and then by GRP and its receptor GRPR. Most MrgprC11^+^ nerve fibers also coexpress other pruritogenic receptors such as H1R for histamine, MrgprA3 for chloroquine, 5-HTR1F for serotonin and IL-4Rα/IL-13Rα1 for IL-4/IL-13. Pruritus by histamine and chloroquine is mediated by NPPB and its receptor NPRA, and then by GRP-GRPR signaling. Chronic pruritus is known to be associated with astrocytosis of the spinal cord. Many abbreviated molecules are listed in Table 1.

In addition to mediating the sensation of pruritus, IL-31 promotes nerve fiber elongation and the branching of murine small-diameter DRG neurons [48,57], which is abrogated in DRG neurons from *Il31ra*-deficient mice [48]. Moreover, the IL-31-induced nerve fiber elongation is independent of TRPV1 but dependent on STAT3 activation [48]. Interestingly, STAT3 also plays a pivotal role in the formation of reactive astrogliosis in the spinal cord, which causes chronic itch [58]. Targeting the STAT3 axis is thus a potential strategy for treating chronic pruritus (Figure 2).

## 4. IL-31 and Keratinocytes

In addition to its pruritogenic function, IL-31 is known to directly inhibit the differentiation of keratinocytes by downregulating the expression of barrier/differentiation-related proteins such as filaggrin, involucrin and cytokeratin 10, which results in the disruption of epidermal barrier function [38,59,60]. IL-31 also increases the expression of IL-1α, IL-20 and IL-24 in keratinocytes and these cytokines are partly responsible for IL-31-mediated downregulation of skin barrier formation [38,60]. In contrast, IL-31 increases the production of antimicrobial peptides such as S100A7 and defensin β4 [60]. Significant amounts of IL-31 are detected in human sweat [20]. Upon stimulation with IL-31, keratinocytes can produce CCL2 [20]. However, the pathophysiological role of IL-31 in keratinocyte biology is not fully understood.

## 5. IL-31 in Skin Diseases with Pruritus

As IL-31 is a potent pruritogenic cytokine, its levels in skin and serum have been examined in various pruritic skin diseases. Increased IL-31 expression was found in lesional and nonlesional skin of patients with atopic dermatitis [24,61,62]. In addition, IL-31-expressing T cells are increased in the lesional skin of atopic dermatitis [62]. Serum levels of IL-31 are well correlated with disease severity in atopic patients [63,64]. Serum levels of IL-31 are elevated in canine atopic dermatitis and are significantly correlated with the pruritus score of afflicted dogs [65].

Prurigo nodularis is associated with severe pruritus. In this condition, the intensity of pruritus is correlated with the numbers of dermal IL-31^+^ cells and dermal IL-31RA^+^ cells [17]. Major cellular sources of dermal IL-31 are T cells and macrophages, while IL-31RA-expressing cells are mostly mast cells and macrophages [17].

The number of IL-31RA^+^ or OSMRβ^+^ cells is also correlated with pruritus intensity in patients with bullous pemphigoid [15]. Most of the dermal cells expressing IL-31 are eosinophils in bullous pemphigoid [15].

The majority of IL-31-expressing cells are CD68^+^ macrophages in stasis dermatitis [16]. The number of IL-31^+^CD68^+^ cells was shown to be significantly increased in stasis dermatitis with severe pruritus compared with that without severe pruritus [16]. These CD68^+^ macrophages also coexpress CD163 and are classified as M2 macrophages [16]. The preferential expression of IL-31 in CD163^+^ M2 macrophages also occurs in scabies infection [14].

Serum levels of IL-31 are significantly elevated in patients with chronic pruritus of unknown origin compared with those in healthy controls [66]. In addition, serum IL-31 levels are significantly increased in patients with chronic urticaria or psoriasis; however, no significant correlation was found between pruritus intensity and IL-31 levels in these diseases [67].

In patients receiving hemodialysis, individuals with uremic pruritus have significantly increased serum levels of IL-31 compared with the levels in those without pruritus [68]. Oweis et al. measured serum levels of IL-31, IL-13 and IL-33 in hemodialysis patients and healthy controls [69]. The serum level of IL-31, but not those of IL-13 and IL-33, was found to be significantly elevated in hemodialysis patients compared with that in healthy controls. However, the serum level of IL-13, but not those of IL-31 and IL-33, was revealed to be significantly correlated with itch intensity [69]. Serum and cutaneous levels of IL-31 are also elevated in cutaneous T-cell lymphoma [19,70,71,72,73] and are described as being correlated with itch intensity in some reports [19,73]. In addition, the serum level of IL-31 is also increased in patients with intrahepatic cholestasis of pregnancy and hepatitis B virus-related liver cirrhosis, which are often associated with pruritus [74,75]. These studies stress the definite or potential role of IL-31 in pruritus induced by diverse skin disorders.

## 6. Control of Pruritus by Targeting IL-31 Signaling

The efficacy of anti-IL-31 treatment against pruritus was first proven in atopic dermatitis (Table 2).

Monotherapy with the anti-IL-31RA antibody nemolizumab was found to significantly attenuate the pruritus in atopic dermatitis compared with placebo control [10,76], for as long as 64 weeks [77]. Concomitant use of nemolizumab further enhanced the anti-pruritic effects of topical steroid [78,79]. Nemolizumab also improved dermatitis severity, sleep disturbance, quality of life and impairment of work productivity/activity [78,79,80]. In addition, the safety profile of nemolizumab is acceptable, without any notable severe adverse events [78,79]. Moreover, in canine atopic dermatitis, anti-canine IL-31 antibody (lokivetmab) significantly inhibited scratching in dogs with canine atopic dermatitis [82,83,84,85]. Proactive treatment with lokivetmab also reduced the flare of canine atopic dermatitis [85].

Recently, nemolizumab has also been reported to attenuate pruritus and skin inflammation associated with prurigo nodularis compared with placebo [11]. In addition, the anti-OSMRβ antibody vixarelimab (KPL-716) is now under a clinical trial for the treatment of prurigo nodularis (ClinicalTrials.gov Identifier: NCT03816891) [81].

## 7. Conclusions

Pruritus is a specific sensation associated with skin inflammation, such as atopic dermatitis and prurigo nodularis. Pruritus is also associated with systemic disorders such as kidney and liver diseases. Since chronic pruritus markedly deteriorates the quality of life of afflicted individuals, its therapeutic control is important. Recent studies have elucidated a close relationship of pruritus with IL-31, which is produced by Th2 cells, macrophages, dendritic cells and eosinophils. In accordance with this, intervention in the binding of IL-31 to its specific receptor IL-31RA/OSMRβ by specific antibodies has been proven to inhibit the pruritus in atopic dermatitis and prurigo nodularis. Targeting the IL-31 signal may thus be a promising strategy for improving the pruritus associated with diverse skin diseases.

## Figures and Tables

**Figure 1 jcm-10-01906-f001:**
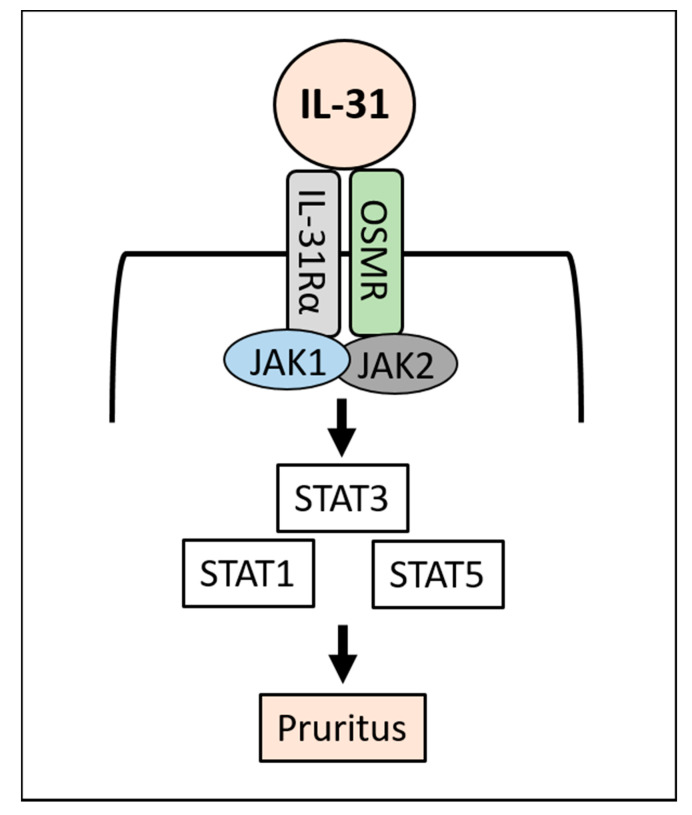
IL-31 signaling.

**Figure 2 jcm-10-01906-f002:**
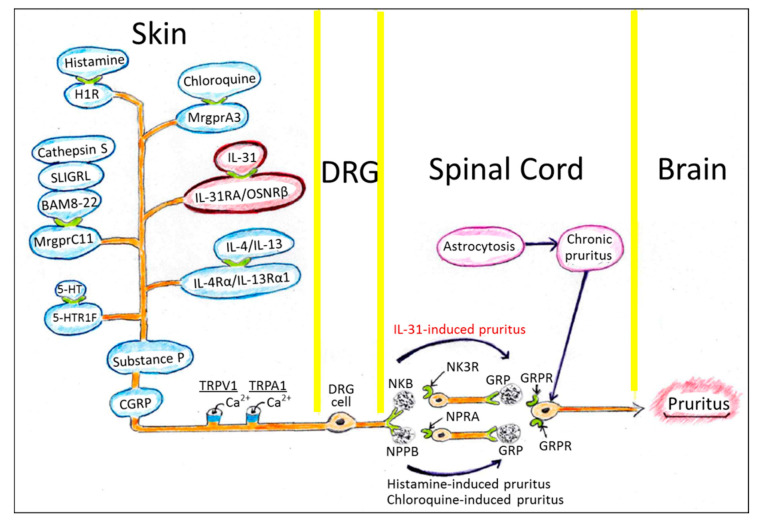
Murine sensory nervous system transmitting the sensation of pruritus.

**Table 1 jcm-10-01906-t001:** Representative pruritogens and their receptors.

Pruritogens	Receptors
Histamine	Histamine H1 receptor (H1R), H4R
BAM8-22, SLIGRL, Cathepsin S	Mas-related G-protein-coupled receptor C11 (MrgprC11)
Chloroquine	MrgprA3
5-Hydroxytriptamine (5-HT), LY344864	5-Hydroxytriptamine receptor 1F (5-HTR1F)
Substance P	Neurokinin 1R (NK1R)
Opioid	μ-Opioid receptor *
Leukotriene B4	Leukotriene B4 receptor 1 (BLT1R)
Endothelin-1	Endothelin receptor type A (ETAR)
SLIGRL, Tryptase, Mucunain, Cathepsin S	Proteinase-activated receptor-2 (PR-2)
AYPGKF, Mucunain, Cathepsin S	PAR-4
IL-31	IL-31 receptor A (IL-31RA)/ Oncostatin M receptor (OSMR)
IL-4/IL-13	IL-4 receptor α (IL-4Rα)/ IL-13 receptor α1 (IL-13Rα1)
Thymic stromal lymphopoietin (TSLP)	TSLP receptor (TSLPR)/ IL-7 receptor α (IL-7Rα)
IL-33	ST2

* κ-Opioid receptor agonists nalfurafine and difelikefalin inhibit μ-opioid receptor.

**Table 2 jcm-10-01906-t002:** Biologics targeting IL-31.

Antibody	Indication	Results
Nemolizumab (anti-IL-31RA) Single injection, [76]	Moderate to severe atopic dermatitis (Phase 1/1b)	Safety profile is tolerable. Significant reduction of pruritus (about −50%) than placebo (about −20%) at week 4.
Nemolizumab (anti-IL-31RA), [10]	Moderate to severe atopic dermatitis (Phase 2)	Significant reduction of pruritus (−43.7% in 0.1 mg/kg, −59.8% in 0.5 mg/kg and −63.1% in 2 mg/kg) than placebo (−20.9%) at week 12.
Nemolizumab (anti-IL-31RA), [77]	Moderate to severe atopic dermatitis (Phase 2) (Long term study)	Significant reduction of pruritus (−73% in 0.1 mg/kg, −89.6% in 0.5 mg/kg and −74.7% in 2 mg/kg) at week 64.
Nemolizumab (anti-IL-31RA), [78]	Moderate to severe atopic dermatitis (Phase 2B) (with topical steroids)	Significant reduction of pruritus (−42.8% in 60 mg/body) by nemolizumab with topical steroid than placebo with topical steroid (−21.4%) at week 16.
Nemolizumab (anti-IL-31RA), [79]	Moderate to severe atopic dermatitis (Phase 2B) (with topical steroids)	Significant reduction of pruritus by nemolizumab with topical steroid (−67.3% in 30 mg/body) than placebo with topical steroid (−35.8%) at week 24.
Nemolizumab (anti-IL-31RA), [80]	Moderate to severe atopic dermatitis (Exploratory analysis)	Significant improvement of work productivity and activity impairment through week 64.
Nemolizumab (anti-IL-31RA), [11]	Moderate to severe prurigo nodularis (phase 2)	Significant reduction of pruritus (−53% in 0.5 mg/kg) than placebo (−20.2%) at week 4.
Vixarelimab (anti-OSMRβ), [81]	Moderate to severe prurigo nodularis (Phase 2a/2b)	In clinical trial

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
