# Peer review of "Interleukin-31 and Pruritic Skin"

_jcm, 2021, doi:10.3390/jcm10091906_

Round 1
Reviewer 1 Report
The article does not add anything new compared to the existing literature: 90 papers on IL 31 and itching of which 30 reviews are already viewable on pubmed. Overall the article is well written and well structured.
Author Response
Reply to the Reviewer 1
The article does not add anything new compared to the existing literature: 90 papers on IL 31 and itching of which 30 reviews are already viewable on pubmed. Overall the article is well written and well structured.
→ Thank you very much for your encouraging and helpful comments. As you mentioned, this is a completely review article but we did our best to describe the updated information. We hope your kind understanding.
Reviewer 2 Report
I read with great interested this review article by Furue and cols. I only have a few minor comments.
- Abstract, line 7. I recommend to avoid mention about pain. We know now that inflammation is related to itch and not necessary pain.
- Introduction, line 21. The use of the term "short-term pruritus" is not well know, and I believe the use "Acute pruritus" would be more appropriate.
- Introduction, line 25. What the authors mean with "miscellaneous eczema"? Please clarified
- Section 2. IL-31 and its signaling. Line 41. IL-31 belongs to the IL-6 or it is related? please clarify
- Figure 1 Substance P is not related to its receptor NK1R, the authors are mentioning this as part of the transmission of itch, or missed the receptor? Please clarify
Author Response
Reply to the Reviewer 2
I read with great interested this review article by Furue and cols. I only have a few minor comments.
→ Thank you very much for your encouraging comments.
- Abstract, line 7. I recommend to avoid mention about pain. We know now that inflammation is related to itch and not necessary pain.
→ Thank you very much for your helpful comment. We agree with you. We delete this sentence in the revised article.
- Introduction, line 21. The use of the term "short-term pruritus" is not well known, and I believe the use "Acute pruritus" would be more appropriate.
→ Thank you very much for your helpful comment. We amended this term to “Acute pruritus”. Thanks again.
- Introduction, line 25. What the authors mean with "miscellaneous eczema"? Please clarified
→ Thank you very much for your helpful comment. We amended this term to “eczema”. Thanks again.
- Section 2. IL-31 and its signaling. Line 41. IL-31 belongs to the IL-6 or it is related? please clarify
→ Thank you very much for your comment. We amended this part to “IL-31 belongs to the “gp130/IL-6 cytokine family”.
- Figure 1 Substance P is not related to its receptor NK1R, the authors are mentioning this as part of the transmission of itch, or missed the receptor? Please clarify
→ Thank you very much for your comment. We understand your question. However, to the best of our knowledge, previous studies clarified that IL-31R+ neurons probably co-express substance P. But it is not still clear whether substance P can activate IL-31R+ neurons or mediated IL-31-induced pruritus.
We hope the revised article is now suitable for publication in JCM.
Reviewer 3 Report
This is a well-written comprehensive review article giving an overview of the role of the IL-31 pathway in chronic itch. I have only one minor suggestion for the authors: authors could consider adding 1 figure summarizing the IL-31 pathway and signaling (section 2 of the manuscript) and 1 table providing an overview of the performed RCTs with antibodides targeting IL-31 including indication and main results.
Author Response
Reply to the Reviewer 3
This is a well-written comprehensive review article giving an overview of the role of the IL-31 pathway in chronic itch. I have only one minor suggestion for the authors: authors could consider adding 1 figure summarizing the IL-31 pathway and signaling (section 2 of the manuscript) and 1 table providing an overview of the performed RCTs with antibodides targeting IL-31 including indication and main results.
→ Thank you very much for your encouraging and helpful comments. According to your comment, we added Figure 1 and Table 2 in the revised article.
We hope the revised article is now suitable for publication in JCM.